# Magnitude and clinical characteristics of cerebral palsy among children in Africa: A systematic review and meta-analysis

**Biruk Beletew Abate**[1]*, **Kindie Mekuria Tegegne**[2], **Alemu Birara Zemariam**[3], **Addis Wondmagegn Alamaw**[4], **Mulat Awoke Kassa**[2], **Tegene Atamenta Kitaw**[5], **Gebremeskel Kibret Abebe**[4], **Molla Azmeraw Bizuayehu**[3]

**1** Assistant Professor in Pediatrics and Child Health Nursing, College of Health Science, Woldia University, Weldiya, Ethiopia, **2** MSc in Psychiatry, College of Health Science, Woldia University, Weldiya, Ethiopia, **3** MSc in Pediatrics and Child Health Nursing, College of Health Science, Woldia University, Weldiya, Ethiopia, **4** MSc in Emergency Medicine and Critical Care Nursing, College of Health Science, Woldia University, Weldiya, Ethiopia, **5** MSc in Adult Health Nursing, College of Health Science, Woldia University, Weldiya, Ethiopia

* birukkelemb@gmail.com

**Data Availability Statement:** All data are available in the manuscript and supporting files.

## Abstract

Cerebral palsy (CP) is the most common motor disability in childhood which causes a child's behavioral, feeding, and sleep difficulties. It remains a poorly studied health problem in Africa. The main aim of this study was assessing the pooled prevalence of Cerebral Palsy (CP) and its clinical characteristics in Africa context. Systematic review and meta-analysis were conducted using Preferred Reporting Items for Systematic Review and Meta-Analysis (PRISMA) guidelines to search articles from electronic databases (Cochrane library, Ovid platform) (Medline, Embase, and Emcare), Google Scholar, CINAHL, PubMed, Maternity and Infant Care Database (MIDIRS). The last search date was on 12/05/ 2023 G. C. A weighted inverse variance random-effects model was used to estimate the pooled estimates of cerebral palsy and its types. The subgroup analysis, publication bias and sensitivity analysis were done. Studies on prevalence and clinical characteristics of cerebral palsy were included. The primary and secondary outcomes were prevalence and clinical characteristics of cerebral palsy respectively. A total of 15 articles with (n = 498406 patients) were included for the final analysis. The pooled prevalence of cerebral palsy in Africa was found to be 3·34 (2·70, 3·98). The most common type is spastic cerebral palsy accounting 69·30% (66·76, 71·83) of all cases. The second one is quadriplegic cerebral palsy which was found to be 41·49% (33·16, 49·81). Ataxic cerebral palsy accounted 5·36% (3·22, 7·50). On the other hand, dyskinetic cerebral palsy was found to be 10.88% (6·26, 15·49). About 32·10% (19·25, 44.95) of cases were bilateral while 25·17% (16·84, 33·50) were unilateral. The incidence of cerebral palsy in Africa surpasses the reported rates in developed nations. Spastic and quadriplegic subtypes emerge as the most frequently observed. It is recommended to channel initiatives toward the strategic focus on preventive measures, early detection strategies, and comprehensive management protocols.

**Funding:** The author(s) received no specific funding for this work.

**Competing interests:** The authors have declared that no competing interests exist.

## Background

Cerebral palsy (CP) is a group of permanent disorders of development that affect different body parts and result in activity limitations with the manner of walking, muscle tone, posture, and coordination of movement. This is attributed to a non-progressive disturbance that occurs in the development of the fetal or infant brain [1]. There are various forms of cerebral palsy, each distinguished by the area of the brain that is injured. This occurs when signals are not correctly sent from the brain to the muscle. Despite the fact that numerous studies show that CP can result from a wide range of conditions, such as early birth, illnesses, injuries, and medical issues, the pathophysiology of this issue is unclear [2]. Cerebral palsy movement abnormalities are frequently accompanied with sensory, perception, cognition, communication, and behavior issues, seizures, and subsequent musculoskeletal problems [3]. Children with CP are present with certain characteristics/types which include spasticity, hypotonia, diplegia, hemiplegia, and dystonia [4]. Usually, motor disorders are more common and accompanied by perception cognition, sensation, communication and behavior due to epilepsy and secondary musculoskeletal problems [1]. CP greatly affects the childhood characteristics such as behavioral difficulties, feeding difficulties, social skills, and sleep [5].

CP is the most common neurologic problem causing motor disability in children [6, 7]. According to the World Health Organization(WHO) report globally, there were 10% of children (approximately 200 million) suffer from physical disability, mental deficiencies or developmental delay, and impaired learning abilities [8]. Furthermore, more than three fourth of the world's disabled population live in low-income countries, many of these in Africa [9]. Recent population-based studies from around the world reported that the prevalence of CP was ranging from 1 to nearly 4 per 1,000 live births [4]. The prevalence of CP is variably reported across different countries which are 1.89 per 1,000 live births in Norway [10], and 2.2 per 1,000 live births in Denmark [11]. Furthermore, a population-based study in Bangladesh reported that 3.4 per 1,000 children [12].

In Africa, the prevalence of CP is also compared to developed nations. Evidently, studies on hospital clinical samples suggest prevalence ranging from 2 to 10 cases per 1000 live births from Egypt, Uganda, South Africa, and South Egypt, respectively [13–18]. Moreover, a population-based study in Uganda revealed that the prevalence of CP was 2.9 per 1,000 children in 2017 [19]. In line with the global trend, CP places a heavy burden of disease on children, families, and society in both developed and developing countries [4, 11]. The previous studies done in different countries showed that several factors were associated with CP. Among these factors, asphyxia at birth, low birth weight, intrauterine infections and multiple gestations were the most important determinants for CP [20–22]. Cases of CP varies in presentation, etiology, evolution, severity, medical and rehabilitation needs, comorbidities, and outcomes [23]. Furthermore, compared to younger ages, older children with cerebral palsy (CP) were substantially less prevalent, and this tendency was reflected in the decline in prevalence. These results implied that a substantial death rate existed among children with cerebral palsy, especially in the most severely impacted children. The paucity of information on the epidemiology of cerebral palsy (CP) in low- and middle-income countries (LMICs) highlights the maltreatment of and shortage of resources for children with CP in these countries, which lowers their survival rate for consecutive birth dates [19]. It might be appreciated at early or later age. Cerebral palsy was diagnosed in 43.4% of children after the first year of life, 32.4% after the second half of life, and 24.1% before the age of six months [24]. This figure shows there is time delay in the diagnosis of CP.

Globally, several strategies and interventions have been tried to reduce the prevalence and adverse sequels of CP among children [7, 9]. It was anticipated that advancements in these

areas would lead to lower rates of cerebral palsy because prenatal events are thought to account for about 75% of all cases of cerebral palsy. These include electronic fetal monitoring, cesarean sections, and generally improving obstetric and neonatal care [4, 8, 25]. Beyond these efforts, the problem has still a public concern, particularly in developing nations [26], because their obstetric and neonatal advanced medical care is limited [27]. Few rigorous population-based studies have recently been published from Uganda [19], and Bangladesh [12] revealing large discrepancy in prevalence between them and studies from high income countries. Large differences in the prevalence of CP were found between studies conducted in high-income countries and those conducted in African nations. These investigations unequivocally showed that the data from these disparate pieces of evidence or from research conducted in wealthy nations, and they suggested the necessity of a thorough evaluation of CP from low- and middle-income nations [28]. There is a lack of structured and consistent screening policy for developmental disabilities amongst children in Africa [18]. Due to this circumstance and delayed presentation, many children with disabilities have gone undiagnosed and consequently have not received the necessary assistance. Consequently, the severity and impact of CP in Africa are understated. Because of stigma, families with disabled children in African nations often find themselves shut out of society. Because they are frequently denied access to the necessities of recognition, education, health care, and socialization, the majority of these children thereafter face numerous social, economic, and political obstacles [29].

In Africa different studies have been conducted regarding the magnitude and clinical characteristics of cerebral palsy, however findings from these small studies lack consistency and results are variable making it challenging to recommend actions. To date, a rigorous systematic review and meta-analysis of the overall prevalence of CP are lacking in resource-poor settings despite the fact that large burden of the disease. Hence, this study is intended to assess the pooled prevalence of CP and its clinical characteristics among children in Africa. The result of this systematic review and meta-analysis will provide a pooled data on the burden of cerebral palsy which can be used as a baseline in designing strategies for prevention and control of cerebral palsy.

## Methods

### Search strategy

This systematic review and meta-analysis review assessed studies that provide data on the magnitude and clinical characteristics of cerebral palsy in the Africa context. We searched these articles from the following databases: Cochrane library, Ovid platform (Medline, Embase, and Emcare), Google Scholar, CINAHL, PubMed, Maternity and Infant Care Database (MIDIRS), and institutional repositories in Africa countries on 12/05/ 2023 G. C. The search in all database included keywords that are the combinations of population, condition/outcome, and context. A snowball searching for the references of relevant papers for linked articles was also performed.

The following search map was applied: (prevalence OR magnitude) AND (Children [MeSH Terms] OR infant OR child OR childhood) AND (cerebral palsy [MeSH Terms] OR "developmental disabilities", OR "neurological impairment," OR "childhood disability") AND (ataxia OR dyskinesia OR spastic) AND ("clinical characteristics [MeSH Terms]" OR type) AND (Africa OR "developing countries") on PubMed database (S1 Table). These search terms were further paired with the names of African countries. Thus, the key searching terms were considering Africa countries that compose of Ethiopia, Djibouti, Somalia, Egypt, Eritrea, Sudan, Tanzania, Kenya, Nigeria, Uganda etc. Using those key terms, we used the Boolean operator "OR" (to connect key terms/phrases within the same concept), "AND" (to connect key terms

/phrases between two concepts), and "NOR" to filter out. In addition, we used truncation (*), adjacency searching (**ADJn)**, and wildcard symbols to find variations in spelling and variant word endings on the Ovid databases. Moreover, we applied relevant limits (filters) such as a limit to human studies only. The sample search strategy for Medline is provided in S1 Table.

## Study selection and screening

The retrieved studies were exported to Endnote version 8 reference managers to remove duplicate studies. Two investigators (BB and KM) independently screened the selected studies using article's title and abstracts before retrieval of full-text papers. We used pre-specified inclusion criteria to further screen the full-text articles. Disagreements were discussed during a consensus meeting with other reviewers (AB and AW) for the fin selection of studies to be included in the systematic review and meta-analysis. The retrieved studies were imported in covidence platform to remove duplicate studies and to do the whole screening process.

## Inclusion and exclusion criteria

Studies that assess the magnitude and clinical characteristics of cerebral palsy among children under the age of 18 were considered. Citations without abstract and/or full text, anonymous reports, editorials, and qualitative studies were excluded from the analysis. The Prevalence of cerebral palsy was considered as the proportion of children with cerebral palsy among 1000 risk population.

## Population

Studies conducted among children under the age of 18 were considered.

## Intervention

Not applicable.

## Study area

Studies conducted in Africa context.

## Study design

All observational studies.

## Outcome

Prevalence of cerebral palsy and its clinical characteristics/type.

## Quality assessment

The authors appraised the quality of the studies by using the Joanna Briggs Institute (JBI) quality appraisal checklist [17, 30]. There was a team of four reviewers and the papers were split amongst the team. Each paper was then assessed by two reviewers and any disagreements were discussed with the third and the fourth reviewers. Studies were considered as low risk or good quality when it scored 4 and above for all designs (cross-sectional, and cohort) [19], whereas the studies scored 3 and below were considered as high risk or poor quality and excluded (S2 Table). Furthermore, we thoroughly extract adjusted confounders and main findings from all included studies (Table 1). Similar methodology has been used in previously published works [17, 18, 31–37].

**Table 1. Distribution of included studies on the prevalence and characteristics of cerebral palsy in Africa, 2023.**

| Sr No | Authors | Year | Country | Study design | Sample size | Prevalence (%) | Characteristics/types of cerebral palsy | | | | | | |
|---|---|---|---|---|---|---|---|---|---|---|---|---|---|
| | | | | | | | Spastic | Quadriplegic | Ataxic | Dyskinestic | Bilateral | Unilateral | Mixed |
| 1. | Labena, F. et al [42] | 2021 | Ethiopia | Cross-sectional | 402 | .. | 88.9 | 62.5 | 1.4 | 7.6 | 9.7 | 27.1 | 2.1 |
| 2. | Kakooza-Mwesige, A. et al [43] | 2016 | Uganda | Cross-sectional | 135 | .. | 23 | | 9.6 | 12.6 | 45.9 | 23.7 | 8.1 |
| 3. | Ngassa, E.A., et al [44] | 2018 | Tanzania | Cross-sectional | 120 | 17.77 | .. | .. | .. | .. | .. | .. | .. |
| 4. | Kakooza-Mwesige, A., et al [19] | 2017 | Uganda | Cross-sectional | 31756 | 2.9 | .. | .. | 2 | 9 | 40 | 46 | 2 |
| 5. | Karumuna, J. et al [45] | 1990 | Tanzania | Cohort | 100 | 10 | .. | .. | 9 | .. | 20 | 15 | 5 |
| 6. | El-Tallawy, HN., et al [46] | 2008 | Egypt | Cross-sectional | 25540 | 2.03 | 65.4 | 42.5 | 3.8 | 3.8 | 9.6 | 13.5 | 26.9 |
| 7. | El-Tallawy, H.N., et al [50] | 2010 | Egypt | Cross-sectional | 25540 | 2.03 | .. | .. | .. | .. | .. | .. | .. |
| 8. | Abas, O. et al [16] | 2017 | Egypt | Cross-sectional | 198776 | 1 | 72.5 | 30.3 | 7 | 16 | 48.27 | 21.4 | |
| 9. | El-Tallawy, H.N., et al [46] | 2014 | Egypt | Cross-sectional | 12788 | 3.6 | 72.5 | | 3.9 | .. | .. | .. | 23.5 |
| 10. | Mangamba, D.K., et al [51] | 2022 | Cameroon | Cross-sectional | 4064 | 4.86 | .. | .. | .. | .. | .. | .. | .. |
| 11. | R Duke, et al. [52] | 2020 | Nigeria | Cross-sectional | 171200 | 2.3 | 69.9 | .. | 9.8 | 4.6 | 60.2 | 39.8 | 8.3 |
| 12. | Tsige, S., et al. [47] | 2021 | Ethiopia | Cross-sectional | 207 | .. | .. | .. | .. | 3.4 | 10.4 | 60.4 | 21.8 | 4 |
| 13. | Coombe, H.J. et al. [48] | 2017 | South Africa | Cross-sectional | 94 | .. | .. | 27.6 | 9.6 | 29.8 | 12.8 | 16 | 2.1 |
| 14. | Salih, K. et al. [49] | 2020 | Sudan | Cohort | 108 | .. | .. | 43.5 | 3.7 | | 13.9 | 25.9 | 2.8 |
| 15. | Couper, J. [14] | 2002 | South Africa | Cross-sectional | 2 036 | 10 | .. | .. | .. | .. | .. | .. | .. |

## Data extraction

The authors developed a data extraction form on the excel sheet and the following data were extracted for eligible studies: year of publication, country, and study design, the definition of cerebral palsy, and clinical characteristics / type of cerebral palsy. The data extraction sheet was piloted using 4 papers randomly, and it was adjusted after piloted the template. Two of the authors (BB and KM) extracted the data using the extraction form in collaboration. The third and fourth (MA and AB) authors checked the correctness of the data independently. Any disagreements between reviewers were resolved through discussions with third and fourth reviewers when required. The mistyping of data was resolved through crosschecking with the included papers.

## Synthesis of results

The authors transformed the data to STATA 17 for analysis after it was extracted in an excel sheet considering prevalence, and type/characteristics reported. We pooled the overall prevalence estimates of cerebral palsy by a random effect meta-analysis model. We examined the heterogeneity of effect size using the Q statistic and the $I^2$ statistics. In this study, the $I^2$ statistic value of zero indicates true homogeneity, whereas the value 25%, 50%, and 75% represented low, moderate and high heterogeneity, respectively [38–41]. Subgroup analysis was done by

the study country, study design, and year of publication. Sensitivity analysis was employed to examine the effect of a single study on the overall estimation. Publication bias was checked by the funnel plot and more objectively through Egger's regression test.

## Results

A total of 5394 studies were identified; 5380 from different databases and 14 from other sources. After duplication removed, a total of 2,431 articles remained (2963 removed by duplication). Finally, 206 studies were screened for full-text review, and 15 articles with (n = 498406 patients) were included for the final analysis (Fig 1, and S2 Table).

### Characteristics of included studies

Fifteen studies were included in this systematic review and meta-analysis [14, 16, 19, 42–52]. Two studies were from Ethiopia [42, 47], two from Uganda [19, 43], two from Tanzania [44, 45], three from Egypt [16, 46, 50], one from Cameroon [51], one from Nigeria [52], two from South Africa [14, 48], and one from Sudan [49] (Table 1).

### Prevalence of cerebral palsy in Africa

Most of the included studies (n = 11) have reported the prevalence of cerebral palsy per 1000 live births [14, 16, 19, 45, 46, 50–52]. The prevalence of cerebral palsy was ranged from 1(95%

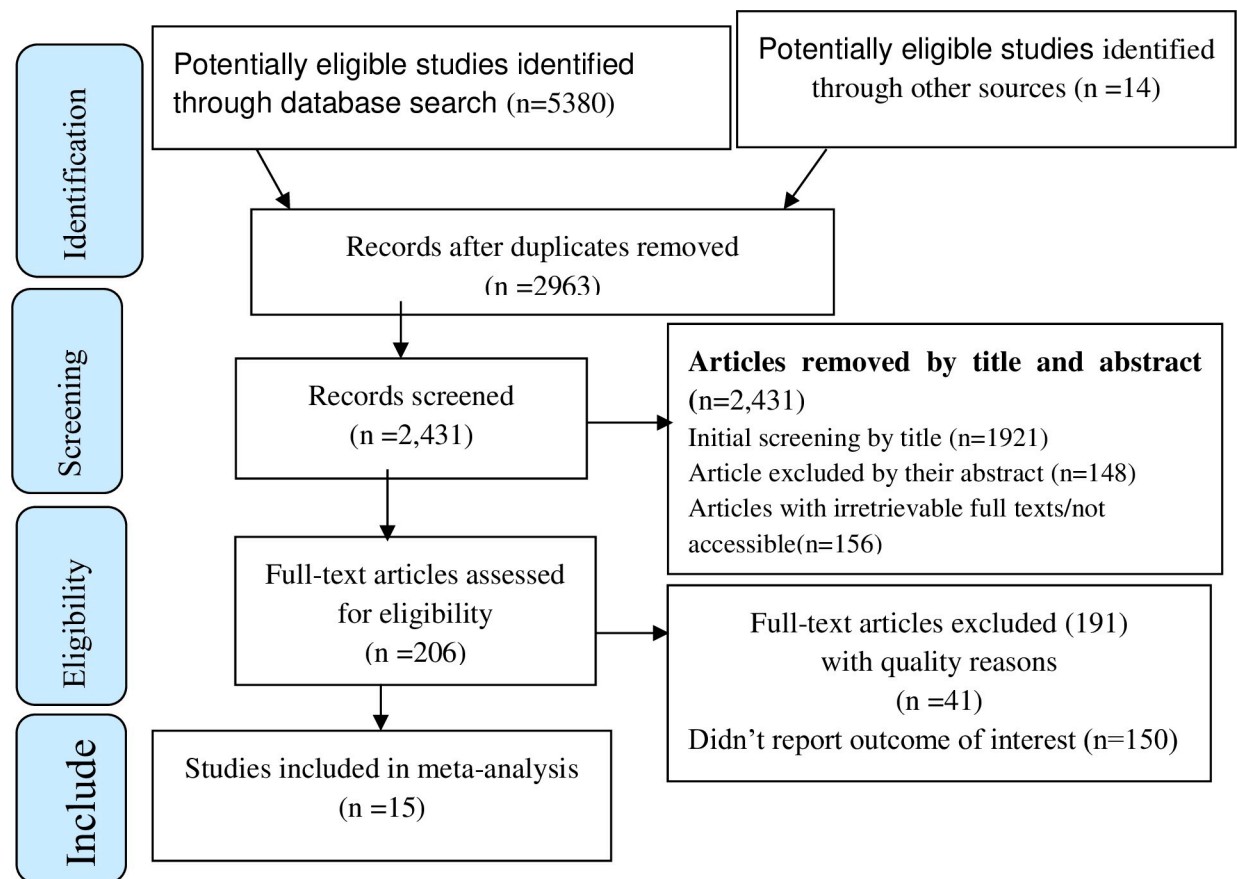

**Fig 1. PRISMA–adapted flow diagram showed the results of the search and reasons for exclusion [79].**

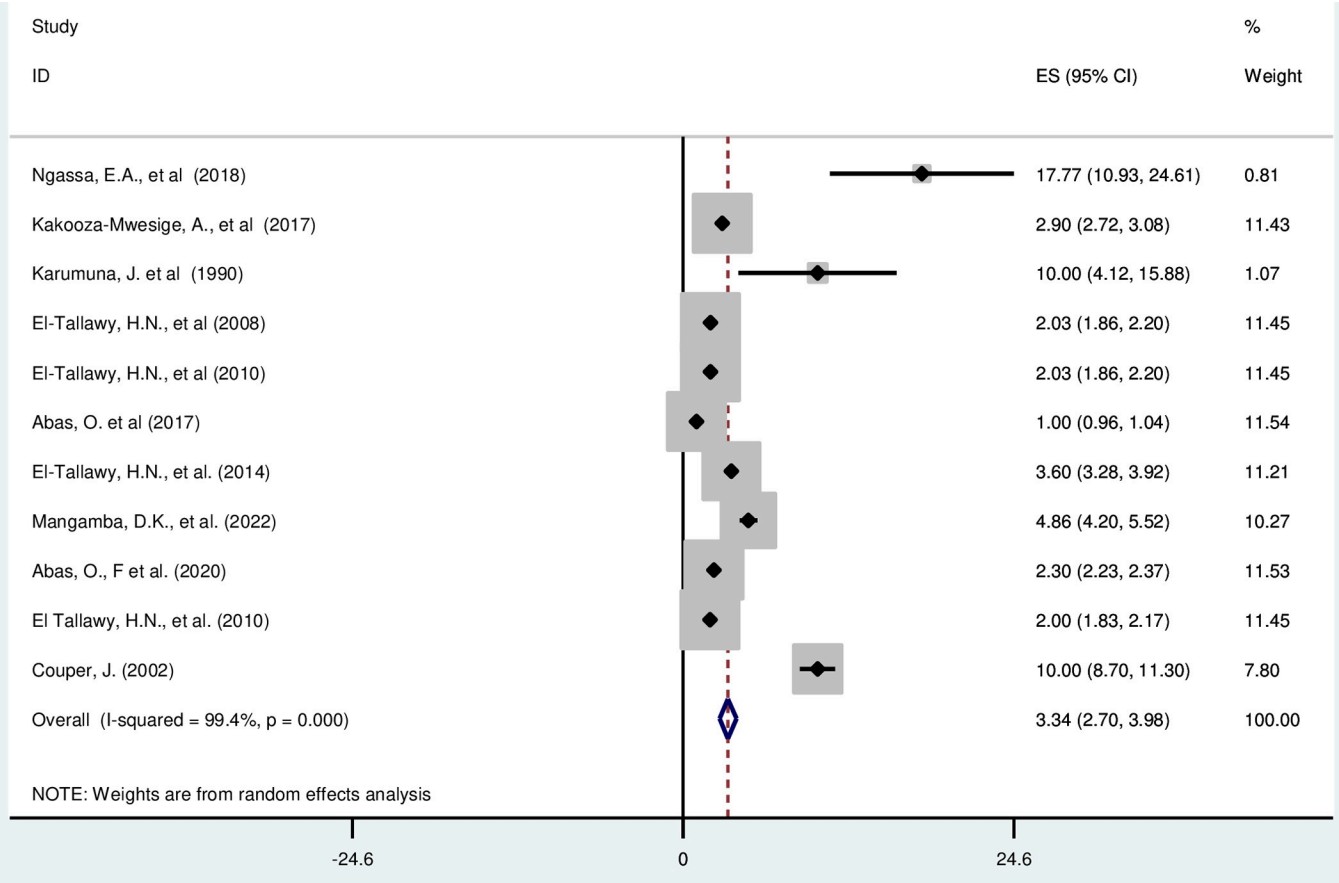

**Fig 2. Forest plot showing the pooled prevalence of cerebral palsy among children in Africa.**

CI: 0·96, 1·04) to 17·77 (95% CI: 10·93, 24·61). The random-effects model analysis from those studies revealed that, the pooled prevalence of cerebral palsy in Africa was found to be 3·34 (95% CI: 2·70, 3·98) (95% CI: $I^2$ = 99·4%; p < 0·001) (Fig 2).

## Subgroup analysis

Subgroup analysis was done through stratified by country, and sample size. Based on this, the prevalence of cerebral palsy was found to be 2·90(2·72–3·08) in Uganda, 13·68(6·08–21·28) in Tanzania, 2·12 (1·37, 2·88) in Egypt, 4·86 (4·20, 5·52), 2·30 (2·23, 2·37) in Nigeria and 10·00 (8·70,11·30) in South Africa (Fig 3).

## Publication bias

A funnel plot showed asymmetrical distribution. The Egger's regression test-value was 0·009, which indicated that, the presence of publication bias. Due to the presence of publication bias (S1 Fig), we employed a trim and fill analysis and one study was added and the prevalence of cerebral palsy becomes 3·193 (S2 Fig).

## Sensitivity analysis

We also employed a leave-one-out sensitivity analysis to identify the potential source of heterogeneity in the analysis of the prevalence of cerebral palsy in Africa. The results of this

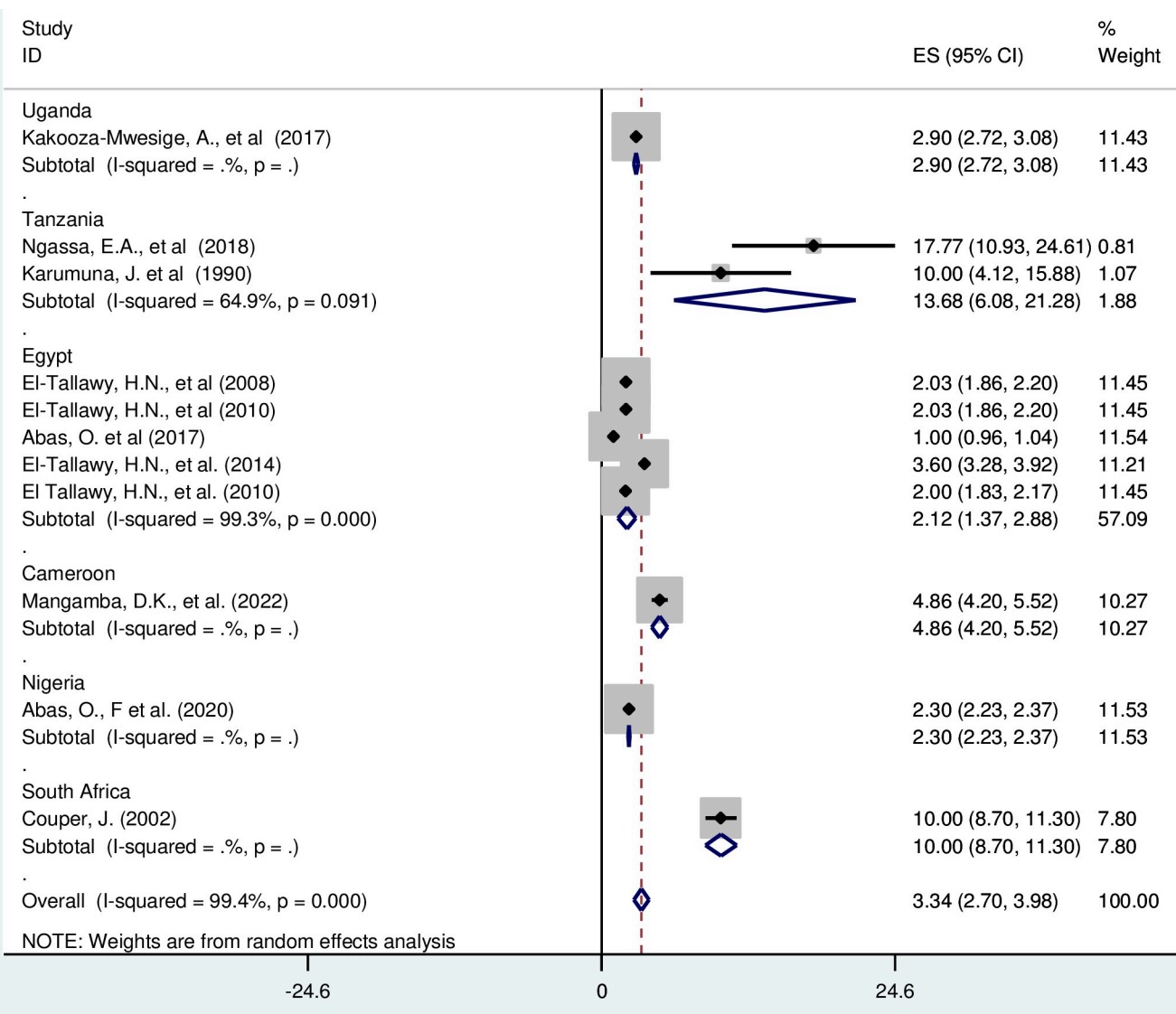

**Fig 3. Forest plot showing subgroup analysis for prevalence of cerebral palsy among children in by countries in Africa.**

sensitivity analysis showed that the findings were not dependent on a single study. Our pooled estimated prevalence of cerebral palsy varied from 2·76 (2·12–3·39) to 3·55 (2·84–4·25) after the deletion of a single study (S3 Fig).

## Clinical characteristics of cerebral palsy in Africa

**Spastic cerebral palsy.** *Pooled prevalence.* Spastic cerebral palsy is the most common type of cerebral palsy in Africa. It is characterized by jerky movements, muscle tightness and joint stiffness. Six of the included studies have reported the magnitude of spastic cerebral palsy in percentage [16, 19, 42, 43, 46, 52]. The prevalence of spastic cerebral palsy was ranged from 23·00%(95% CI: 15·90, 30·10) to 88·90 (95% CI: 85·83, 91·97). The random-effects model analysis from those studies revealed that, the pooled prevalence of spastic cerebral palsy in Africa was found to be 69·30% (95% CI: 66·76, 71·83) (95% CI: $I^2$ = 99·5%; p < 0·001) (Fig 4).

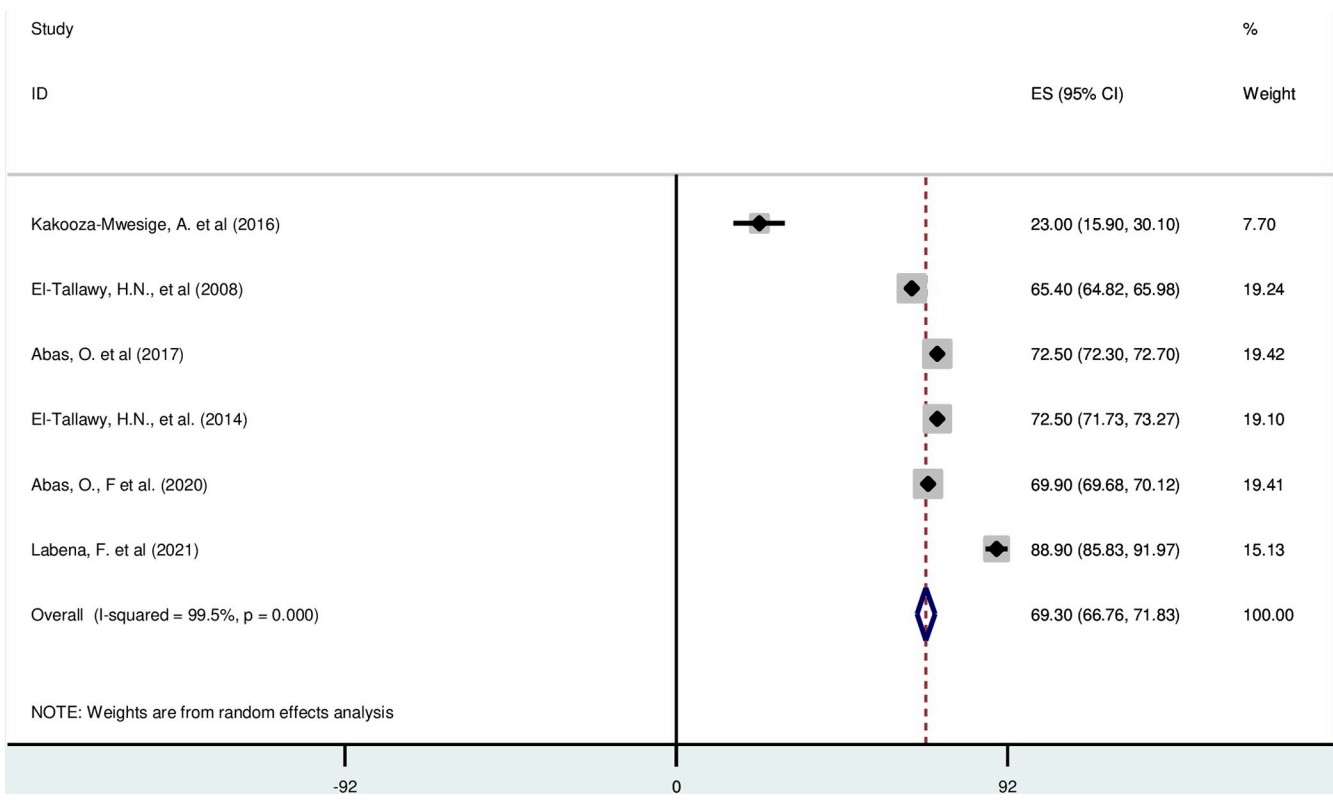

**Fig 4. Forest plot showing the pooled prevalence of spastic cerebral palsy among children in Africa.**

*Subgroup analysis.* The subgroup analysis was done through stratified by country, and sample size. Based on this, the prevalence of spastic cerebral palsy was found to be 23·00%(95% CI: 15·90, 30·10) in Uganda, 70·13% (65·54,74·72) in Egypt, 69·90%(69·68, 70·12) in Nigeria, and 88·9%(85·83, 91·97) in Ethiopia (S4 Fig).

*Publication bias.* A funnel plot showed symmetrical distribution. The Egger's regression test-value was 0·468, which indicated that, the absence of publication bias. As a result, we didn't conduct trim and fill analysis (S5 Fig).

**Quadriplegic cerebral palsy in Africa.** *Pooled prevalence.* Dyskinetic CP is characterized by involuntary, uncontrolled, and recurring movements with fluctuating muscle tone [3, 28]. Most of the included studies (n = 5) have reported the magnitude of quadriplegic cerebral palsy in percentage from total cerebral palsy cases [16, 19, 42, 46, 48, 49]. The prevalence of unilateral cerebral palsy was ranged from 27.60% (95% CI: 18.56, 36.64) to 62·50 (95% CI: 57·77, 67·23). The random-effects model analysis from those studies revealed that, the pooled prevalence of quadriplegic cerebral palsy in Africa was found to be 41·49% (95% CI: 33·16, 49·81) (95% CI: $I^2$ = 99·7%; p < 0·001) (Fig 5).

*Subgroup analysis.* The subgroup analysis was done through stratified by country. Based on this, the prevalence of quadriplegic unilateral cerebral palsy was found to be 36·40% (24·44–48·35) in Egypt, 62·5% (57·77, 67·23) in Ethiopia, 27·60% (18·56, 36·64) in South Africa, and 43·50% (34·15, 52·85) in Sudan (S6 Fig).

*Publication bias.* A funnel plot showed symmetrical distribution. The Egger's regression test-value was 0·360, which indicated that, the absence of publication bias. As a result, we didn't conduct trim and fill analysis (S7 Fig).

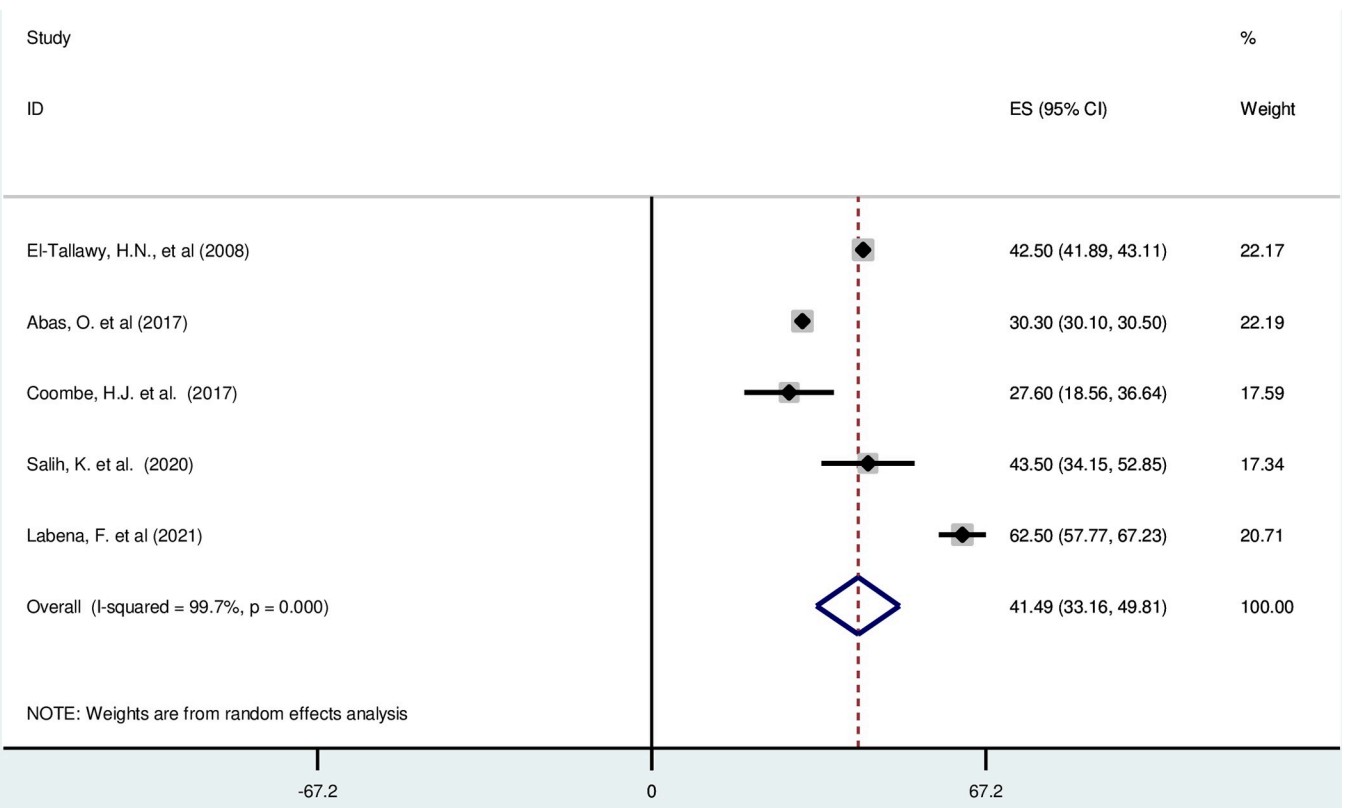

**Fig 5. Forest plot showing the pooled prevalence of quadriplegic cerebral palsy among children in Africa.**

**Ataxic cerebral palsy in Africa.** *Pooled prevalence.* Ataxic cerebral palsy characterized by trouble with balance and coordination [53]. Most of the included studies (n = 11) have reported the prevalence of ataxic cerebral palsy per 1000 [16, 19, 42, 43, 45–49, 52]. The prevalence of cerebral palsy was ranged from 1·4(95% CI: 0·25, 2·55) to 9·80 (95% CI: 9·66, 9·94). The random-effects model analysis from those studies revealed that, the pooled prevalence of cerebral palsy in Africa was found to be 5·36 (95% CI: 3·22, 7·50) (95% CI: $I^2$ = 99·8%; p < 0·001) (Fig 6).

*Subgroup analysis.* The subgroup analysis was done through stratsified by country, and sample size. Based on this, the prevalence of ataxic cerebral palsy was found to be 5·38(2·02–12·78) in Uganda, 9·00(3·39–14·61) in Tanzania, 4·90 (2·46, 7·35) in Egypt, 2·09 (0·23, 3·96) in Ethiopia, 9·60 (3·64, 15·56) in South Africa and 10·00 (8·70,11·30) in Sudan (S8 Fig).

*Publication bias.* A funnel plot showed symmetrical distribution. The Egger's regression test-value was 0·633, which indicated that, the absence of publication bias. Due to the absence of publication bias, we did not employ a trim and fill analysis (Fig 7).

**Dyskinestic cerebral palsy in Africa.** *Pooled prevalence.* Dyskinetic cerebral palsy is characterized by dystonia, athetosis, and chorea [54]. Most of the included studies (n = 8) have reported the magnitude of ataxic cerebral palsy [16, 19, 42, 43, 46–48, 52]. The prevalence of ataxic cerebral palsy was ranged from 3·80% (95% CI: 3·57, 4·03) to 29·80 (95% CI: 20·55, 39·05). The random-effects model analysis from those studies revealed that, the pooled prevalence of unilateral cerebral palsy in Africa was found to be 10·88% (95% CI: 6·26, 15·49) (95% CI: $I^2$ = 100%; p < 0.001) (Fig 8).

*Subgroup analysis.* The subgroup analysis was done through stratified by country, and sample size. Based on this, the prevalence of unilateral cerebral palsy was found to be 9·67%(6·92–

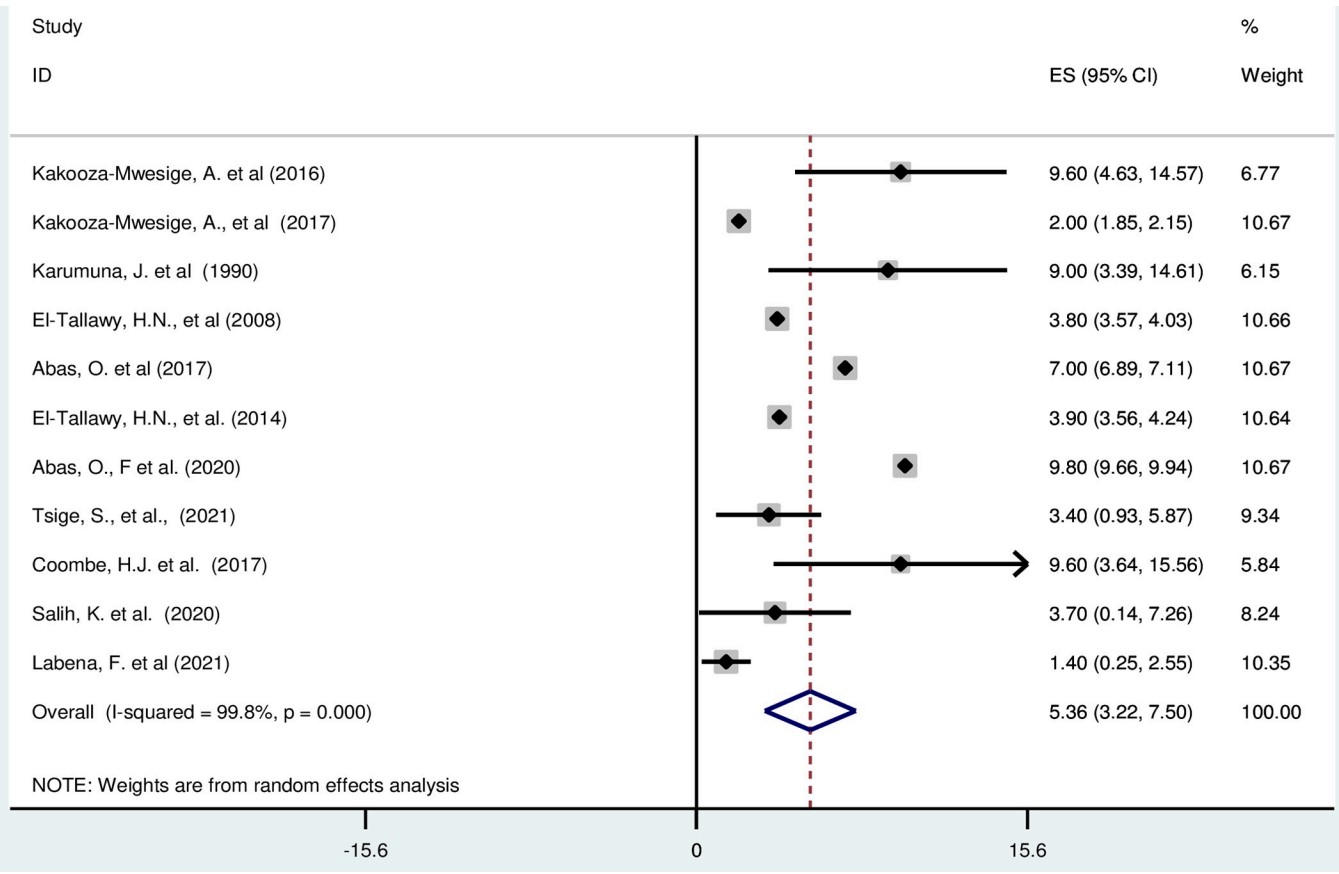

**Fig 6. Forest plot showing the pooled prevalence of ataxic cerebral palsy among children in Africa.**

12·42) in Uganda, 9·90% (2·06, 21·86) in Egypt, 4·60%(4·50, 4·70) in Nigeria, 8·51%(5·94, 11·06) in Ethiopia, and 29·80% (20·55, 39·05) in South Africa (S10 Fig).

*Publication bias.* A funnel plot showed symmetrical distribution. The Egger's regression test-value was 0·728, which indicated that, the absence of publication bias. As a result, we didn't conduct trim and fill analysis (S11 Fig).

**Bilateral cerebral palsy in Africa.** *Pooled prevalence.* Bilateral cerebral palsy is a problem with movement, co-ordination and development which affects both sides of the body [55]. Most of the included studies (n = 10) have reported the magnitude of bilateral cerebral palsy in percentage from total cerebral palsy cases [16, 19, 42, 43, 46–48, 52]. The prevalence of unilateral cerebral palsy was ranged from 9·60% (95% CI: 9·24, 9·96) to 60·40 (95% CI: 53·74, 67·06). The random-effects model analysis from those studies revealed that, the pooled prevalence of bilateral cerebral palsy in Africa was found to be 32·10% (95% CI: 19·25, 44·95) (95% CI: $I^2$ = 100%; p < 0·001) (Fig 9).

*Subgroup analysis.* The subgroup analysis was done through stratified by country, and sample size. Based on this, the prevalence of unilateral cerebral palsy was found to be 41·40% (36·48–46·31) in Uganda, 20·00% (12·16–27·84) in Tanzania, 28·94% (8·96, 66·83) in Egypt, 60·20%(59·97, 60·43) in Nigeria, 34·96 (14·73, 84·4) in Ethiopia, 16·00% (8·59, 23·41) in South Africa, and 13·9% (7·38, 20·42) in Sudan (S12 Fig).

*Publication bias.* A funnel plot showed symmetrical distribution. The Egger's regression test-value was 0·529, which indicated that, the absence of publication bias. As a result, we didn't conduct trim and fill analysis (S13 Fig).

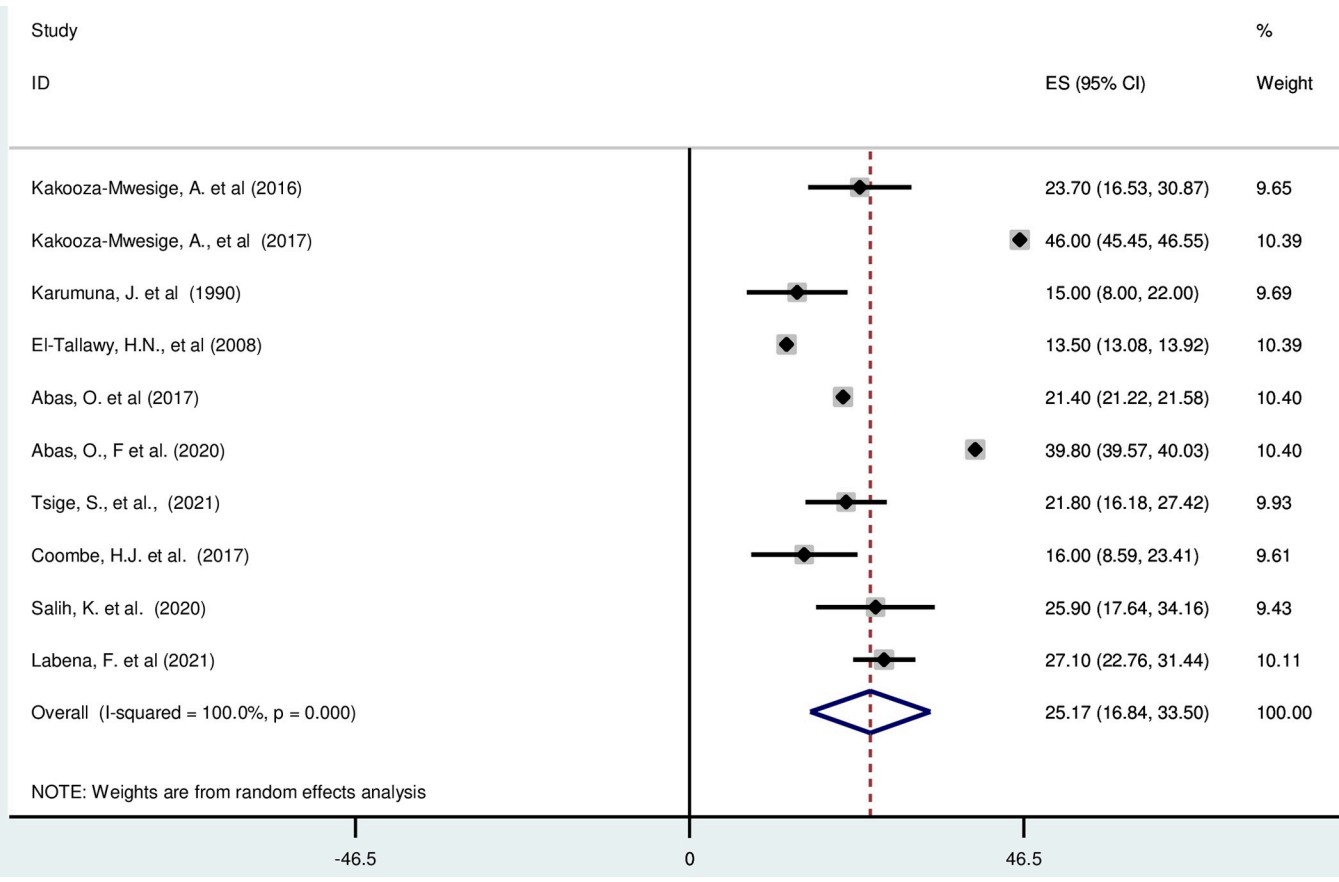

**Fig 7. Forest plot showing the pooled prevalence of unilateral cerebral palsy among children in Africa.**

**Unilateral cerebral palsy in Africa.**   *Pooled prevalence*. Unilateral Cerebral Palsy is characterized by hemiplegia and hemiparesis, is a condition that affects muscle control and function on one side of the body [56]. Most of the included studies (n = 10) have reported the magnitude of unilateral cerebral palsy in percentage from total cerebral palsy cases [16, 19, 42, 43, 45–49, 52]. The prevalence of unilateral cerebral palsy was ranged from 13·50% (95% CI: 13·08, 13·92) to 23·70 (95% CI: 16·53, 30·87). The random-effects model analysis from those studies revealed that, the pooled prevalence of unilateral cerebral palsy in Africa was found to be 25·17% (95% CI: 16·84, 33·50) (95% CI: $I^2$ = 100%; p < 0·001) (Fig 4).

*Subgroup analysis*. The subgroup analysis was done through stratified by country. Based on this, the prevalence of unilateral cerebral palsy was found to be 35·15%(13·30–56·99) in Uganda, 15%(8·00–22·00) in Tanzania, 17·45% (9·71, 25·19) in Egypt, 39·8%(39·57, 40·03) in Nigeria and 10·00 (8·70,11·30), 24·76%(19·61, 29· 92) in Ethiopia, 16·00% (8·59,23·41) in South Africa, and 25·9% (17·64, 34·16) in Sudan (S14 Fig).

*Publication bias*. A funnel plot showed symmetrical distribution. The Egger's regression test-value was 0·903, which indicated that, the absence of publication bias. As a result, we didn't conduct trim and fill analysis (S15 Fig).

**Mixed type cerebral palsy in Africa.**   *Pooled prevalence*. Mixed cerebral palsy occurs when a child exhibits symptoms of more than one type of cerebral palsy [57]. Most of the included studies (n = 10) have reported the magnitude of mixed cerebral palsy in percentage [19, 42, 43, 45–49, 52]. The prevalence of mixed cerebral palsy in Africa was ranged from to

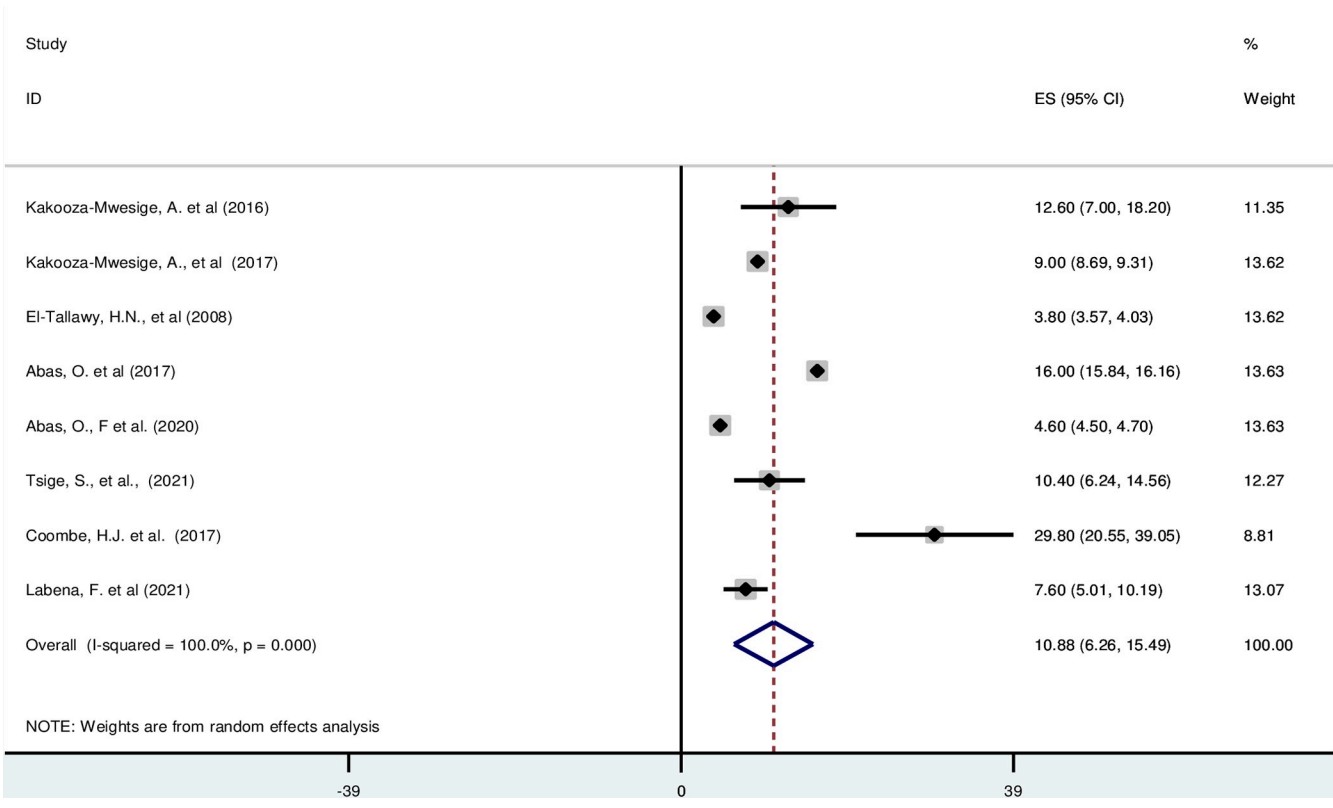

**Fig 8. Forest plot showing the pooled prevalence of dyskinestic cerebral palsy among children in Africa.**

2·00%(1·85, 2·15) to 26·9% (95% CI: 26·36, 27·44). The random-effects model analysis from those studies revealed that, the pooled prevalence of mixed cerebral palsy in Africa was found to be 8·58% (95% CI: 4·06, 13·11) (95% CI: $I^2$ = 99·9%; p < 0·001) (Fig 10).

*Subgroup analysis.* The subgroup analysis was done through stratified by country. Based on this, the prevalence of unilateral cerebral palsy was found to be 4·60%(1·31–10·51) in Uganda, 5·0%(0·73–9·27) in Tanzania, 25·21% (21·88, 28·54) in Egypt, 8·3%(8·17, 8·43) in Nigeria and 2·70 (0·97, 4·42) in Ethiopia, 2·10% (0·80, 5·00) in South Africa, and 2·80% (0·31, 6·59) in Sudan (S16 Fig).

*Publication bias.* A funnel plot showed symmetrical distribution. The Egger's regression test-value was 0·468, which indicated that, the absence of publication bias. As a result, we didn't conduct trim and fill analysis (S17 Fig).

## Discussion

The most prevalent form of motor disability that affects children is cerebral palsy (CP). The brain may malfunction during fetal or prenatal development due to insufficient oxygenation, or it may result from damage to the developing infant's brain during the postpartum period. In this systematic review and meta-analysis, we aimed to assess the magnitude and clinical characteristics of CP among children in Africa. Accordingly, the pooled prevalence of cerebral palsy was found to be 3·3 per 1000 live births (2·69,3·92). This finding is in line with the study done in USA [58]. This might be because African descendants were surveyed in that study. However, this finding is higher than CP prevalence in most other countries in United State and Europe which is estimated to be 2–2·5 of 1000 [59–61]. Compared to some studies

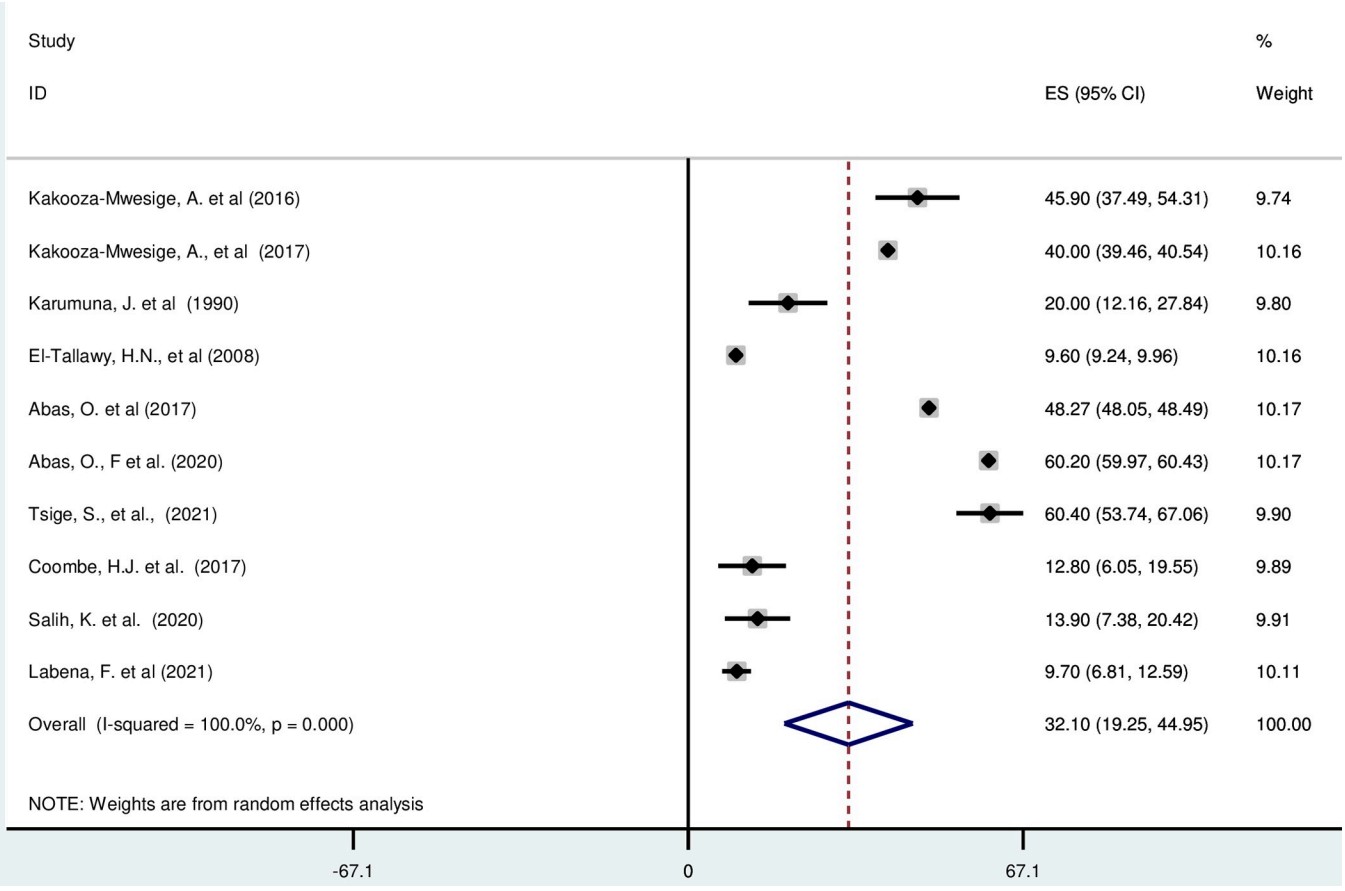

**Fig 9. Forest plot showing the pooled prevalence of bilateral cerebral palsy among children in Africa.**

conducted in Arabic-speaking countries, 1·8 [7], Europe, 2·25 [62], India, 2·27 [63], Canada, 2·57 [64], and Japan 1·88 [65], the pooled prevalence of cerebral palsy is greater in Africa. This discrepancy might be because of the high magnitude of perinatal complications such as birth asphyxia and neonatal infections in Africa [66]. In addition, this might be due to the fact that in Africa, there are challenges to manage cerebral palsy due to: inaccessibility of basic care, limited availability of diagnostic facilities, limited number of trained and expertise personal in managing cerebral palsy and exacerbated by lack of appropriate intervention, medication, surgical procedure and regular physical care [67]. In fact, these improper managements might have contributed to the prevalence of cerebral palsy. Furthermore, prematurity, low birthweight, kernicterus, asphyxia and neonatal infections are among the main causes of cerebral palsy, and they are more common in Africa than in western nations [67]. This implies there may be a greater proportion of children with more severe disability in resource-poor countries because of delayed presentation of a range of disorders and absence of early intervention services. Moreover, the clinical spectrum of CP differs from that of affluent countries in resource-poor, developing nations. The discrepancy probably explained by the multifactorial causes of CP and difference in quality of health care delivery between the two regions, low- and middle-income countries and developed countries [19, 68]. The burden difference might also be due to absence or inadequacy of inter-disciplinary team approach in developing countries [24]. On the other hand, most studies done in developing countries were institution based and at referral hospitals. Since referral hospitals receive CP patients from different district hospitals, health

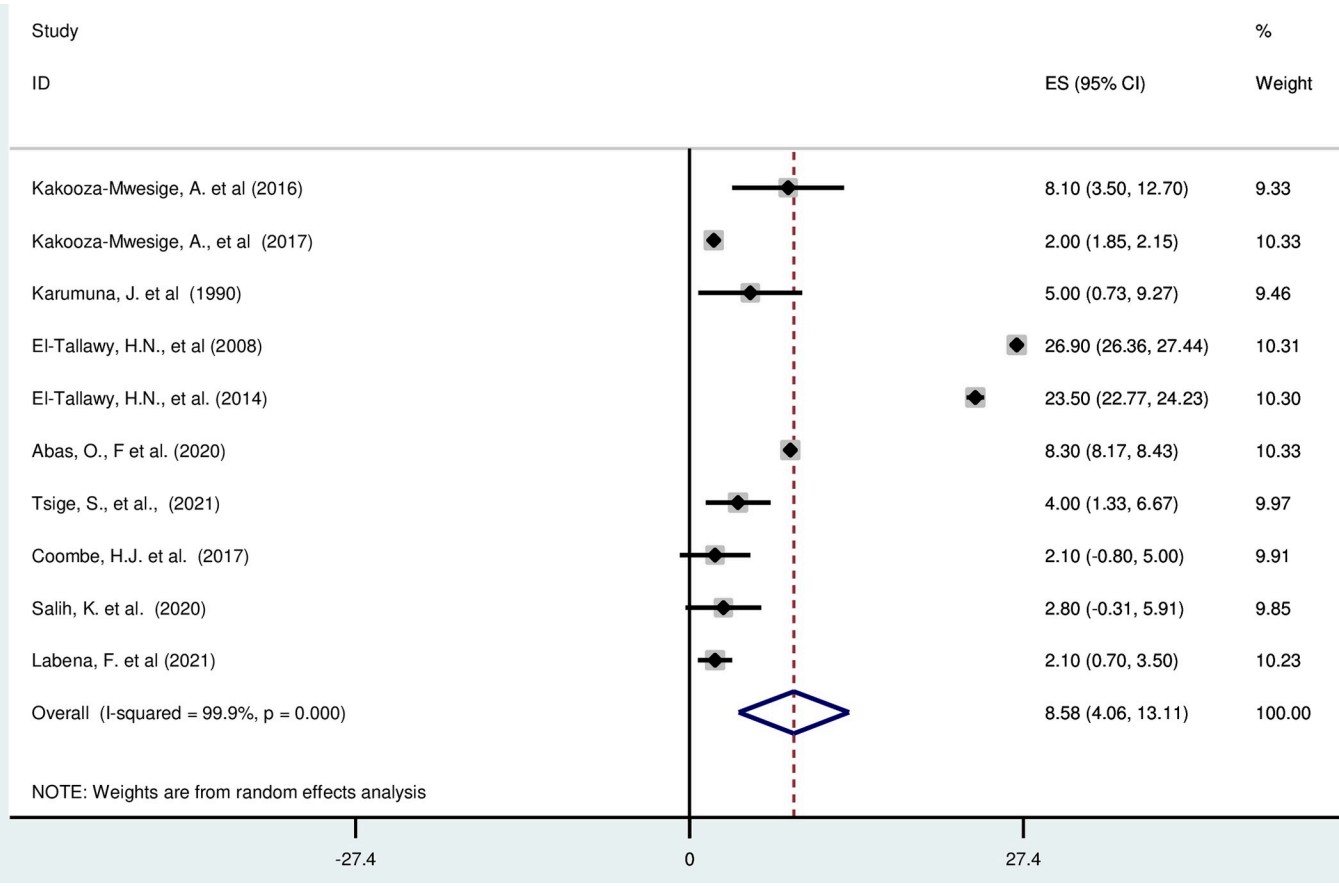

**Fig 10. Forest plot showing the pooled prevalence of mixed type cerebral palsy among children in Africa.**

center and regional hospitals where there are few experts and facilities hence the number of CP patients' data might be aggregated to the referral hospitals [44, 69]. In addition, the high prevalence may be due to improvement in health seeking behavior of the community in developing countries where previously people in Africa believed that having CP patient at home is a curse and tried to hide them at their homes. Tanzania had a higher prevalence of cerebral palsy, at 13·68/1000 (6·08–21·28) compared to studies from other African countries. This might be due to the small sample size included in studies from Tanzania. It is also higher than that of other countries study, Norway 2·1 [70], Thailand 1·0 [71]. This difference might be advance technology used, utilization of advance treatment modalities, regular follow up of antenatal care and sample size difference in other countries like Norway and Thailand.

The results of the current review revealed that the spastic type of cerebral palsy affected the majority of children in Africa which accounts 63·4% of all form cerebral palsy. This finding is in line with different studies done in other regions such as north east Italy, Bangladesh and Nepal [58, 72, 73]. During assessment of perinatal factors, studies identified that those children with spastic subtype of CP had higher rate of fetal distress and PROM during the perinatal period, higher rate of language and speech difficulty and worse functional impairment while those with Dyskinetic and Ataxic CP were found to have higher rate of precipitated labor and deep jaundice during the neonatal period [74]. These findings may indicate that spastic subtype of CP might be related with the high rate of perinatal hypoxic insult in Africa as in cases of fetal distress, while dyskinetic and ataxic forms may be

associated with bleeding and injuries to the deep grey matters of the brain that can happen in cases of precipitated labor. Children with severe forms of CP were shown to have lower levels of communication function, according to a study from Sweden that demonstrated the relationship between communication function and gross and fine motor and cognitive function [75]. Numerous severe (quadriplegic) bilateral spastic type of CP arises from injuries to the full-term brain due to complications during the birth process such as birth asphyxia or acquired central nervous system infections like meningitis [76]. Study done in Misurata Hospital -LIBYA around 50% of cases had malnutrition and anemia [77]. In Greece study, nearly one-third of patients suffered from iron deficiency anemia [78]. Children with disabilities and their families in African countries are frequently excluded from society because of stigmatization. Most of these children consequently confront many challenges socially, economically, and politically because they are often denied the basics of health care, education, socialization, and recognition.

This systematic review and meta-analysis have limitations. There is variation between African countries in terms of culture, the economic profile, political stability, research funding and infrastructure, and health care systems. This makes it very difficult to generalize the results of this review across all countries. Besides, although we have tried to extensively search articles from all countries in Africa, we able to found 15 articles with from few countries; this may also affect the generalizability of the pooled findings. Furthermore, differences in methodology, including hospital versus community sittings, in the included studies also contributed to high levels of heterogeneity. Despite these limitations, these meta-analyses provide a picture of CP and its types among children in Africa.

## Conclusions and recommendations

The overall prevalence of cerebral palsy in Africa found to be high (3·34 /1000) (95% CI: 2·70, 3·98). Compared to findings from developed countries spastic, and quadriplegic cerebral palsy were found to be the most common with pooled prevalence of 69·30% (95% CI: 66·76, 71·83) and 41·49% (95% CI: 33·16, 49·81) respectively. Governmental and non-governmental organizations should target their effort on the prevention, early detection and management of cerebral palsy in Africa. Large multicenter studies, qualitative or mixed-methods studies assessing patient and family understanding of CP, access to resources, and barriers to care are virtually absent in Africa to address recent evidence gap. In addition, there is a need for longitudinal studies that would assess outcomes over time for patients with CP as well as for randomized controlled trials of community-based treatment strategies that would be appropriate in an African setting.

## Supporting information

**S1 Table. Search strategy used for one of the databases.**
(XLSX)

**S2 Table. Quality appraisal result of included studies in Africa, using Joanna Briggs Institute (JBI) quality appraisal checklist [16].**
(XLSX)

**S1 Fig. Shows test of publication bias for prevalence of cerebral palsy among children in by sample size in Africa.**
(TIF)

**S2 Fig. Trim and fill analysis for prevalence of cerebral palsy among children in by sample size in Africa.**
(TIF)

**S3 Fig. Sensitivity analysis for prevalence of cerebral palsy among children in by sample size in Africa.**
(TIF)

**S4 Fig. Subgroup analysis for prevalence of spastic cerebral palsy among children by countries in Africa.**
(TIF)

**S5 Fig. Publication bias for prevalence of spastic cerebral palsy among children in Africa.**
(TIF)

**S6 Fig. Subgroup analysis for prevalence of quadriplegic cerebral palsy among children in Africa.**
(TIF)

**S7 Fig. Publication bias for prevalence of quadriplegic cerebral palsy among children in Africa.**
(TIF)

**S8 Fig. Subgroup analysis for prevalence of ataxic cerebral palsy among children in Africa.**
(TIF)

**S9 Fig. Publication bias for prevalence of ataxic cerebral palsy among children in Africa.**
(TIF)

**S10 Fig. Subgroup analysis for prevalence of dyskinetic cerebral palsy among children in Africa.**
(TIF)

**S11 Fig. Publication bias for prevalence of dyskinetic cerebral palsy among children in Africa.**
(TIF)

**S12 Fig. Subgroup analysis for prevalence of bilateral cerebral palsy among children in Africa.**
(TIF)

**S13 Fig. Publication bias for prevalence of bilateral cerebral palsy among children in Africa.**
(TIF)

**S14 Fig. Subgroup analysis for prevalence of unilateral cerebral palsy among children in Africa.**
(TIF)

**S15 Fig. Publication bias assessment for prevalence of unilateral cerebral palsy among children in Africa.**
(TIF)

**S16 Fig. Subgroup analysis for prevalence of mixed type cerebral palsy among children in Africa.**
(TIF)

**S17 Fig. Publication bias for prevalence of mixed type cerebral palsy among children in Africa.**
(TIF)

**S18 Fig.**
(TIF)

## Author Contributions

**Conceptualization:** Biruk Beletew Abate, Kindie Mekuria Tegegne, Alemu Birara Zemariam, Addis Wondmagegn Alamaw, Mulat Awoke Kassa, Tegene Atamenta Kitaw, Gebremeskel Kibret Abebe, Molla Azmeraw Bizuayehu.

**Data curation:** Biruk Beletew Abate, Kindie Mekuria Tegegne, Alemu Birara Zemariam, Addis Wondmagegn Alamaw, Tegene Atamenta Kitaw, Gebremeskel Kibret Abebe, Molla Azmeraw Bizuayehu.

**Formal analysis:** Biruk Beletew Abate, Kindie Mekuria Tegegne, Alemu Birara Zemariam, Addis Wondmagegn Alamaw, Mulat Awoke Kassa, Tegene Atamenta Kitaw, Gebremeskel Kibret Abebe, Molla Azmeraw Bizuayehu.

**Funding acquisition:** Biruk Beletew Abate, Kindie Mekuria Tegegne, Alemu Birara Zemariam, Addis Wondmagegn Alamaw, Gebremeskel Kibret Abebe, Molla Azmeraw Bizuayehu.

**Investigation:** Biruk Beletew Abate, Kindie Mekuria Tegegne, Alemu Birara Zemariam, Addis Wondmagegn Alamaw, Gebremeskel Kibret Abebe, Molla Azmeraw Bizuayehu.

**Methodology:** Biruk Beletew Abate, Kindie Mekuria Tegegne, Alemu Birara Zemariam, Addis Wondmagegn Alamaw, Mulat Awoke Kassa, Tegene Atamenta Kitaw, Gebremeskel Kibret Abebe, Molla Azmeraw Bizuayehu.

**Project administration:** Biruk Beletew Abate, Kindie Mekuria Tegegne, Alemu Birara Zemariam, Addis Wondmagegn Alamaw, Gebremeskel Kibret Abebe, Molla Azmeraw Bizuayehu.

**Resources:** Biruk Beletew Abate, Kindie Mekuria Tegegne, Alemu Birara Zemariam, Addis Wondmagegn Alamaw, Gebremeskel Kibret Abebe, Molla Azmeraw Bizuayehu.

**Software:** Biruk Beletew Abate, Kindie Mekuria Tegegne, Alemu Birara Zemariam, Addis Wondmagegn Alamaw, Mulat Awoke Kassa, Gebremeskel Kibret Abebe, Molla Azmeraw Bizuayehu.

**Supervision:** Biruk Beletew Abate, Kindie Mekuria Tegegne, Alemu Birara Zemariam, Addis Wondmagegn Alamaw, Tegene Atamenta Kitaw, Gebremeskel Kibret Abebe, Molla Azmeraw Bizuayehu.

**Validation:** Biruk Beletew Abate, Kindie Mekuria Tegegne, Alemu Birara Zemariam, Addis Wondmagegn Alamaw, Tegene Atamenta Kitaw, Gebremeskel Kibret Abebe, Molla Azmeraw Bizuayehu.

**Visualization:** Biruk Beletew Abate, Kindie Mekuria Tegegne, Alemu Birara Zemariam, Addis Wondmagegn Alamaw, Mulat Awoke Kassa, Tegene Atamenta Kitaw, Gebremeskel Kibret Abebe, Molla Azmeraw Bizuayehu.

**Writing – original draft:** Biruk Beletew Abate, Kindie Mekuria Tegegne, Alemu Birara Zemariam, Addis Wondmagegn Alamaw, Mulat Awoke Kassa, Tegene Atamenta Kitaw, Gebremeskel Kibret Abebe, Molla Azmeraw Bizuayehu.

**Writing – review & editing:** Biruk Beletew Abate, Kindie Mekuria Tegegne, Alemu Birara Zemariam, Addis Wondmagegn Alamaw, Mulat Awoke Kassa, Tegene Atamenta Kitaw, Gebremeskel Kibret Abebe, Molla Azmeraw Bizuayehu.

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
