## [Decision Letter · Decision Letter 0]

20 Feb 2024

PGPH-D-23-02244

Magnitude and clinical characteristics of cerebral palsy among children in Africa: A systematic review and Meta-analysis

Dear Dr Abate,

Thank you for submitting your manuscript to PLOS Global Public Health. After careful consideration, we feel that it has merit but does not fully meet PLOS Global Public Health’s publication criteria as it currently stands. Therefore, we invite you to submit a revised version of the manuscript that addresses the points raised during the review process.

We look forward to receiving your revised manuscript.

Kind regards,

Shobhana Nagraj

Academic Editor

Journal Requirements:

1. We noticed you have some minor occurrence of overlapping text with the following previous publication(s), which needs to be addressed:

-http://dx.doi.org/10.1136/bmjopen-2020-039200

-http://dx.doi.org/10.1016/j.spen.2014.01.001

-https://doi.org/10.3390/brainsci12070859

In your revision ensure you cite all your sources (including your own works), and quote or rephrase any duplicated text outside the methods section. Further consideration is dependent on these concerns being addressed.

2. Please provide separate figure files in .tif or .eps format only and remove any figures embedded in your manuscript file. Please also ensure all files are under our size limit of 10MB.

4. We notice that your supplementary [figures/tables] are included in the manuscript file. Please remove them and upload them with the file type 'Supporting Information'. Please ensure that each Supporting Information file has a legend listed in the manuscript after the references list.

Additional Editor Comments (if provided):

Dear Authors,

Thank you for submitting your manuscript to our journal. Please note Reviewer 1's feedback, which recommends a major revision, with detailed feedback and outlining the necessary changes required for the manuscript. We look forward to receiving the revised draft.

Yours sincerely,

Reviewers' comments:

Reviewer's Responses to Questions

**Comments to the Author**

1. Does this manuscript meet PLOS Global Public Health’s publication criteria? Is the manuscript technically sound, and do the data support the conclusions? The manuscript must describe methodologically and ethically rigorous research with conclusions that are appropriately drawn based on the data presented.

Reviewer #1: Yes

Reviewer #2: Yes

2. Has the statistical analysis been performed appropriately and rigorously?

Reviewer #1: Yes

Reviewer #2: Yes

3. Have the authors made all data underlying the findings in their manuscript fully available (please refer to the Data Availability Statement at the start of the manuscript PDF file)?

Reviewer #1: Yes

Reviewer #2: Yes

4. Is the manuscript presented in an intelligible fashion and written in standard English?

Reviewer #1: No

Reviewer #2: Yes

5. Review Comments to the Author

Reviewer #1: Review Report

Title: Magnitude and clinical characteristics of cerebral palsy among children in Africa: A systematic review and Meta-analysis

Date: December 29,2023

General comment

Dear editor, thank you for considering me to review this manuscript. This paper requires a lot of editorial work to improve the logical flow of information and coherence in the description of methods, results of the study, discussion, and recommendation. The draft also requires the correction of grammatical errors. Furthermore, too many results were reported, which makes it difficult to follow and understand. Reporting the characteristics of cerebral palsy at once seems ideal than separately writing them.

There are also too many figures and supplementary materials. This is again the result of the wide result section.

Generalizing the findings is also the main concern here. With only 15 studies included from a few countries, how can we draw inferences for all African countries?

Specific comment

Abstract

1. Background and objective are not well articulated

2. What are the criteria to consider prevalence and clinical characteristics of cerebral palsy as primary and secondary outcome?

3. Clearly present the information about the types of cerebral palsy. It lacks logical flow of the information to be presented.

4. Consider the keywords

5. I’m not sure if the PLOS Global Public Health requires the Strengths and limitations of this study to written before background. I think this is a manuscript prepared for journals like BMJ Open. Follow the guideline of the journal.

Background

1. Abbreviation/acrimony issues

2. Add the significance of the reviews finding for concerned stakeholders

Method

1. Why you’re interested to present the search strategy report of PubMed database only. It is written as…. Thus, the PubMed search combines #1 AND #2 AND #3 AND #4 AND #5. This should be removed.

2. Inclusion and exclusion criteria; specify PICO

3. Why is it important to do subgroup analysis using study design? What value does it add for concerned stakeholders?

Results

1. Improve the quality of PRISMA –flow diagram

2. Under Characteristics of included studies; remove this statement…. Table 1: summarizes the characteristics of the 15 included studies in this systematic review and meta-analysis… and further information that can summarize the information written in table 1

3. Consider clearing out extraneous lines from tables.

4. Think about removing or clipping unnecessary shade/borders from figures, as this affects quality after publication.

5. The country-by-country subgroup study yielded very detailed results. Just list the highest and lowest prevalence; figure 3 presents the results in great detail.

6. The caption for the figure displaying subgroup analysis for the prevalence of cerebral palsy among children in African nations was written figure 3, however it is cited in the paper as Supplementary Fig. 1. This may affect the whole sup. Files order.

7. How important is it to perform subgroup analysis for every cerebral palsy characteristic? Just a small number of studies were used in this particular study. This may have an impact on the subgroup analysis's outcome. You included too many figures in the manuscript by writing a detailed report on the findings and providing extra information. For each kind of cerebral palsy, reports included pooled prevalence, subgroup analysis, publication bias, and sensitivity analysis. There is too much information here, making it difficult to interpret the outcome.

Reviewer #2: he author successfully presented the article in a scientific, clear, and concise approach, demonstrating a profound understanding of complex concepts while effectively communicating them to the target audience

6. PLOS authors have the option to publish the peer review history of their article (what does this mean?). If published, this will include your full peer review and any attached files.

**Do you want your identity to be public for this peer review?** For information about this choice, including consent withdrawal, please see our Privacy Policy.

Reviewer #1: No

Reviewer #2: **Yes: **Oluwayemi Joshua Bamikole

---

## [Decision Letter · Decision Letter 1]

23 May 2024

Magnitude and clinical characteristics of cerebral palsy among children in Africa: A systematic review and Meta-analysis

PGPH-D-23-02244R1

Dear Mr. Abate,

We are pleased to inform you that your manuscript 'Magnitude and clinical characteristics of cerebral palsy among children in Africa: A systematic review and Meta-analysis' has been provisionally accepted for publication in PLOS Global Public Health.

Best regards,

Julia Robinson

Executive Editor

Reviewer Comments (if any, and for reference):

Reviewer's Responses to Questions

**Comments to the Author**

1. If the authors have adequately addressed your comments raised in a previous round of review and you feel that this manuscript is now acceptable for publication, you may indicate that here to bypass the “Comments to the Author” section, enter your conflict of interest statement in the “Confidential to Editor” section, and submit your "Accept" recommendation.

Reviewer #1: All comments have been addressed

2. Does this manuscript meet PLOS Global Public Health’s publication criteria? Is the manuscript technically sound, and do the data support the conclusions? The manuscript must describe methodologically and ethically rigorous research with conclusions that are appropriately drawn based on the data presented.

Reviewer #1: Yes

3. Has the statistical analysis been performed appropriately and rigorously?

Reviewer #1: Yes

4. Have the authors made all data underlying the findings in their manuscript fully available (please refer to the Data Availability Statement at the start of the manuscript PDF file)?

Reviewer #1: Yes

5. Is the manuscript presented in an intelligible fashion and written in standard English?

Reviewer #1: Yes

6. Review Comments to the Author

Reviewer #1: Having thoroughly examined each line of the document, I appreciate the authors for their impeccable writing in every section. I endorse the publication of this research due to its valuable contribution in providing information on the field.

7. PLOS authors have the option to publish the peer review history of their article (what does this mean?). If published, this will include your full peer review and any attached files.

**Do you want your identity to be public for this peer review?** For information about this choice, including consent withdrawal, please see our Privacy Policy.

Reviewer #1: No
